# Extent of illicit cigarette market from single stick sales in Ghana: findings from a cross-sectional survey

Arti Singh ![ORCID] ,[1] Hana Ross,[2] Fiona Dobbie ![ORCID] ,[3,4] Allen Gallagher,[5] Tarja Kinnunen,[1] Divine Darlington Logo,[6,7] Olivia A Boateng,[8] Anna Gilmore,[9] Linda Bauld ![ORCID] ,[10] Ellis Owusu-Dabo[11]

For numbered affiliations see end of article.

**Correspondence to**
Dr Arti Singh;
artisingh_uk@yahoo.com

## ABSTRACT

**Objective** This study aims to measure the extent of illicit cigarette consumption from single stick sales, to determine the nature and types of illicit cigarettes present in Ghana, and to identify the factors associated with illicit cigarette consumption in Ghana.

**Design** A cross-sectional study using empty cigarette packs generated by 1 day's single stick cigarette sales collected from cigarette vendors.

**Setting** Five large cities (Accra, Kumasi, Takoradi, Tamale and Bolgatanga) and three border towns (Aflao, Paga/Hamele and Elubo) in the northern, middle and coastal belt of Ghana.

**Procedure and participants** Ten areas were randomly selected in each city/town, and all shops selling cigarettes within 1 km of the central point were surveyed.

**Outcome measures** (1) Estimates of the share of illicit cigarette packs in the total cigarette sales from vendors selling single stick cigarettes in Ghana; (2) nature and types of illicit cigarette packs; (3) factors associated with illicit cigarette sales in Ghana.

**Results** Of a total of 4461 packs, about 20% (95% CI 18.3 to 20.7) were found to be illicit. Aflao (Ghana-Togo border) and Tamale (Ghana-Burkina Faso border) had the highest percentage of illicit cigarette sales at 99% and 46%, respectively (p<0.001). Over half of the illicit packs originated from Togo (51%), followed by Nigeria (15%) and then Cote d'Ivoire (10%). Adjusted and unadjusted logistic regression models indicated that convenience stores, border towns, pack price and the northern zone had higher odds of illicit cigarette sales.

**Conclusion** To effectively tackle illicit cigarettes, market surveillance and strengthening supply chain control are required, particularly at the border towns and the northern region of the country.

## INTRODUCTION

Illicit tobacco trade continues to remain a threat to global tobacco control efforts. While tobacco consumption is decreasing globally, rapid population growth, increased advertising by the tobacco industry (TI) and growing tobacco consumption among young people in Africa may result in increased number of smokers in the region.[1] Furthermore, the availability and accessibility of cheap, illicit tobacco products are particularly attractive to the region's most vulnerable young population and low-income smokers.[2]

Illicit trade of tobacco products is a major public health problem as lower prices of illicit cigarettes lead to increased cigarette consumption.[3] Despite the difficulties in measuring the extent of illicit tobacco in the market, available estimates indicate that it was about 11.6% worldwide in 2007 and almost 10% in 2015,[3] and these figures are higher for low and middle-income countries (LMICs) including those in the African Region. In response to the threat posed by illicit tobacco trade, the WHO FCTC Protocol to Eliminate Illicit Trade in Tobacco Products (hereby referred to as 'the Protocol') entered into force in 2018.[4] This Protocol gives countries an opportunity to prevent tobacco-related morbidity and mortality by enhancing tobacco supply chain control. Countries that ratify the Protocol commit themselves to adopting a variety of measures, including track and trace systems to prevent and counter illicit trade.

Ghana, one of the first countries to ratify the WHO's Framework Convention on Tobacco Control in 2004, has made some significant progress in tobacco control such as introducing an early advertising ban (1982), the passage of the Tobacco Control Act (in 2012), banning of single stick sales (2017), introduction of mandatory graphic health warnings

## STRENGTHS AND LIMITATIONS OF THIS STUDY

⇒ The empty pack survey required little time and resources to conduct.
⇒ Some retailers may not provide all the packs over the last 24 hours, which could underestimate our findings.
⇒ The study was limited to five cities and three border towns and is not representative of illicit cigarette sales in Ghana as a whole.

(2018) and tax stamps on tobacco products (2018) and more recently the ratification of the Protocol in October 2021.[5] Despite this progress, cigarettes continue to remain cheap and affordable in Ghana.[1] For instance, the price of a pack of the most commonly sold brand of cigarette in Ghana is less than one USD. Although, Ghana does not have an active TI (British American Tobacco (BAT) ceased its local production in 2006), BAT continues to dominate sales of cigarettes and remains the dominant importer of cigarettes into the country via its manufacturing sites in Ibadan and Zaria in Nigeria.[5] The distribution networks of Ghana's leading tobacco companies are well organised in Ghana's major urban cities including Greater Accra, Takoradi, Kumasi and Tamale. Tobacco products including cigarettes in Ghana are mostly sold at unlicensed and unregulated points of sale such as traditional grocery retailers (also known as convenience or provision stores), street vendors, kiosks and drinking bars.[6]

An important challenge that exists in many African countries, including Ghana, is that most governments do not measure the size of illicit tobacco market nor analyse its features on a regular basis. To fully benefit from the Protocol, policymakers seek to connect its normative guidance with empirical data and analysis on countries' illicit tobacco trade. In light of the TI's use of illicit trade to oppose tobacco control measures such as tax increases,[7] it is important to understand the scope and nature of the illicit tobacco trade. To date, there have been no scientific studies to estimate the size of the illicit cigarette market in Ghana.[1] The only available estimates are those produced by the Euromonitor that reports an illicit cigarette market accounting for 39% of total cigarette sales in 2018 (up from 35% in 2017).[8] Estimates by Euromonitor have been criticised for being unreliable and inconsistent, and for lacking independence due to Euromonitor entering into business contracts with Philip Morris International.[5] The objectives of this study were to measure the extent of the illicit cigarette market in selected border and non-border towns in Ghana using an empty pack survey method from single stick sales. The study also assessed the nature and types of illicit cigarettes present in Ghana including the factors associated with illicit cigarette sales in Ghana.

## METHODS
### Study sites
A cross-sectional study was conducted during the months of August 2020 to January 2021 in five major cities in Ghana (Accra, Tamale, Kumasi, Takoradi and Bolgatanga) and three border towns (Aflao, Paga/Hamele and Elubu) across the three zones of Ghana (Northern, Middle and Coastal). These districts were selected to represent socioeconomic, cultural and geographical diversity.

### Research design
A modified approach based on the analysis of empty cigarette packs collected directly from retailers was used.

This method was adapted from similar studies in India[9] and Bangladesh[10] and is particularly useful in countries where single stick sales are a common practice. Within each large city or border town, 10 smaller geographical areas were selected using Ghana Post Codes. A central point (such as a government building, market place or taxi station) was determined in each of them for retailor pack collection. A team of four research assistants and a coordinator walked 1 km along both sides of a busy street (0.5 km forward and 0.5 km back) starting from the central point to identify tobacco retailers. All retailers identified were provided with verbal and written information about the study and requested to sign a consent form if they agreed to participate. Following consent being obtained, an empty bag with a unique identifier was given to retailers and they were asked to deposit all cigarette packs emptied throughout the day as a result of single sticks of cigarette sales in the bag provided. The bags were collected back from the retailers at the end of a 24-hour period and retailers were given a small monetary incentive (up to a maximum amount of USD 10). Consenting retailers also participated in a 20–25 min survey on illicit cigarette sales, common brands and pricing of cigarettes sold each day. Pack prices were recorded for each of the 10 and 20 stick packs. The sample size equation to obtain the minimum number of packs collected from each selected city/town was adapted from a toolkit for measuring illicit tobacco in LMICs.[11] We obtained a minimum sample size of 2600 packs, assuming prevalence of illicit cigarette sales of 25%, with 95% level of confidence and margin of error of 0.15.

### Classification of packs
Empty cigarette packs were cleaned and assigned unique ID and were analysed and their characteristics recorded. Pack data included the brand name, country of origin, the presence of graphical and/or textual health warnings, the language of the warning, the pack size (10/20 stick pack) and compliance of these warning messages with existing packaging requirement for Ghana. A conservative definition to classify an illicit cigarette pack in Ghana according to the Food and Drugs Authority (FDA), the regulatory body and the focal point for tobacco control in Ghana,[12] includes at least one of the following attributes:
a. Absence of authentic tax stamps.
b. Absence of textual and pictorial warnings (current pack warnings in Ghana are required to be a combined picture and text health warning in English to cover 50% of the front principal display area and 60% of the back principal display area of the pack, positioned in the lower portion).[13]
c. Absence of the inscription 'FOR SALE IN GHANA ONLY' displayed on the side panel of the product pack.
d. Health warnings not in English.

Trained research assistants evaluated tax stamp authenticity using the tax stamp mobile application developed by the Ghana Revenue Service.[14]

## Analysis

Data were first entered into excel, cleaned and analysed via R studio V.1.4.1717. There was missing information from three of the pack data and these were removed from the final analysis. The unit of analysis was each cigarette pack. Continuous variables such as price/pack were changed to categorical (low and high price category) for 2–7 Ghana cedis and 8–14 GHC, respectively (1USD=6 GHC) for measures of association and continuous for the regression analysis. Descriptive information was reported as frequencies and percentages for city, country zone (northern, middle and coastal zones), retail shop type (drinking bars, convenience stores and kiosks), border and non-border towns, country of origin (based on the inscription on the packs on sale restricted to respective country, eg, for sale in Togo only or Nigeria, etc) and illicit and licit cigarette. Pack characteristics such as pictorial health warning (absent/present), textual health warning (absent/present), warning labels in English (absent/present), tax stamps (absent/present) and *for sale in Ghana* sign (absent/present) were captured. The relationship between illicit tobacco and the categorical variables (city type, country zone, type of shops, border and non-border town, price/packs, cigarette brand and country of origin) were first studied using $\chi^2$ or Fisher's exact test (when the number in the table was <6). Due to the binary nature of the outcome variable (licit/illicit), simple and multiple logistic regression was performed to evaluate the unadjusted and adjusted predictive values of the potential confounding variables, respectively, based on the existing literature[15][16] (figure 1). Subsequently, a cluster analysis was performed to identify the effect of vendors on the sale of illicit single e stick sales in Ghana.

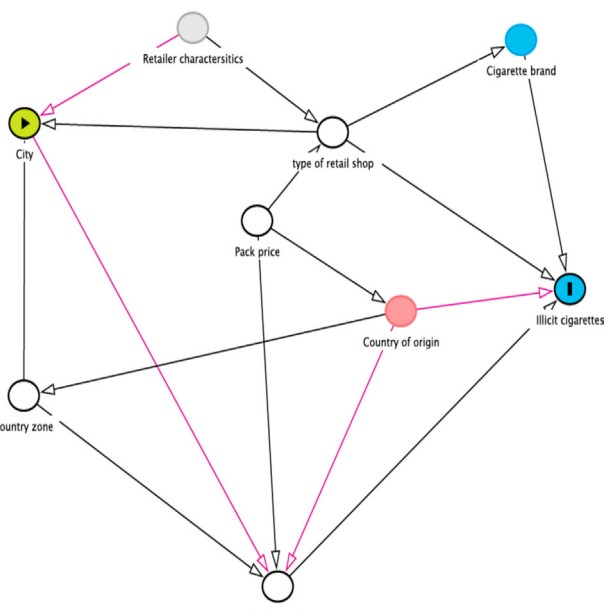

**Figure 1** Causal diagram of illicit cigarette consumption from single stick sales in Ghana (potential confounders were border towns, country zone, pack prices and type of retail shop).

The results are presented as ORs with a 95% CI, with significance set at an alpha level of 5% (p≤0.05).

### Patient and public involvement

No patients and/or the public were involved in the design, or conduct, or reporting or dissemination plans of this research.

## RESULTS

A total of 425 retailers were approached for the study, of whom 384 (90%) consented to collect packs and participate in the survey. An average of 12 cigarette packs were collected from single stick sales in a 24-hour period. A total of 4461 packs were collected from 384 retailers in the selected cities and towns. All retailers (100%) in the study sold single sticks. A total of 871 out of 4461 (20%, 95% CI 18.34 to 20.66) packs were classified as illicit based on the criteria for classification approved by the FDA. A third (31%) of the packs collected from the northern zone of Ghana were illicit and almost 7 out of 10 (69%) packs from the border towns were illicit. Almost all the packs collected from Aflao (Ghana-Togo border) were illicit (99%), followed by Tamale (46%) and the Paga/Hamele (Ghana-Burkina Faso border) (27%) and Elubu (21%) (Ghana—Cote d'Ivoire border) (table 1). In terms of the retail selling points, 3 out of 10 (29%) packs collected from convenience stores were illicit, followed by drinking bars (18%) (p<0.001). Over 60% of the packs collected within the price category of 2–7 GHC were illicit. The most common brand of cigarettes sold in Ghana is Rothmans Kingsize, London Brown/White and Pall Mall (figure 2).

Of all the 871 illicit packs collected, the most common brands of single stick sales were from Business Royal (24%), followed by Fine (21%) and Oris (12%).

All packs from 555 and London Brown/White (manufactured by BAT) were licit (100%) (table 1).

For the classification of illicit packs, majority were characterised by absence of tax stamps (94%), *for sale in Ghana* sign (92%), warning labels not in English (77%) and absence of text and pictorial warning labels (28%).

Almost all the packs collected were the 20-stick pack (98%). The average price/pack of the 20-stick packs was 8.5 GHC and that for 10-stick was 3.3 GHC. Illicit packs had an average price/pack of 5.4 GHC (SD 1.5, range 2–12 GHC) while licit pack was 9.1 GHC (SD 2.1, range 2–14 GHC). Close to half of the illicit packs originated from Togo (51%), followed by Nigeria (15%) and then Cote d'Ivoire (10 %). About 2% of packs that were destined for Ghana were classified as illicit as the packs did not conform to the current labelling requirements as approved by FDA.

Table 2 shows the results from adjusted and unadjusted logistic regression of the factors associated with illicit cigarette sales in Ghana. The odds of illicit cigarette sales were 1.8-folds and 3.5-folds higher in convenience stores as compared with drinking bars in the unadjusted

**Table 1**  Determinants of illicit cigarette sale in Ghana

| | Illicit cigarette packs (n=871) | Licit cigarettes packs (n=3590) | Total |
|---|---|---|---|
| *Country zone* | | | |
| Northern | 368 (30.6) | 835 (69.4) | 1203 (100) |
| Middle | 8 (1.2) | 656 (98.8) | 664 (100) |
| Coastal (south) | 495 (19.1) | 2099 (80.9) | 2594 (100) |
| *P-value** | *<0.001* | | |
| *Border/non-border* | | | |
| Border | 493 (68.5) | 227 (31.5) | 720 (100) |
| Non-border | 378 (10.1) | 3363 (89.1) | 3741 (100) |
| *P-value** | *<0.001* | | |
| *City/town (border/non-border)* | | | |
| Accra (non-border) | 17 (1.5) | 1147 (98.5) | 1164 (100) |
| Kumasi (non-border) | 8 (1.2) | 651 (98.8) | 659 (100) |
| Takoradi (non-border) | 1 (0.1) | 767 (99.9) | 768 (100) |
| Bolgatanga (non-border) | 7 (1.8) | 390 (98.2) | 397 (100) |
| Tamale (non-border) | 345 (45.8) | 408 (54.2) | 753 (100) |
| Elubu (Cote d'ivoire border) | 44 (21.1) | 165 (78.9) | 209 (100) |
| Paga/Hamele (Burkina Faso border) | 16 (26.6) | 42 (72.4) | 58 (100) |
| Aflao (Togo border) | 433 (98.6) | 20 (1.4) | 453 (100) |
| P-value* | <0.001 | | |
| *Shop type* | | | |
| Drinking bar | 477 (18.2) | 2139 (81.8) | 2616 (100) |
| Kiosks | 31 (5.2) | 563 (94.8) | 594 (100) |
| Convenience stores | 363 (29.0) | 888 (71.0) | 1251 (100) |
| *P-value** | *<0.001* | | |
| *Price/pack (GHC)* | | | |
| Low price[2–7] | 778 (61.2) | 494 (38.8) | 1272 (100) |
| High price[8–14] | 93 (2.9) | 3096 (97.1) | 3189 (100) |
| P-value* | <0.001 | | |
| *Cigarette brand (manufacturer)* | | | |
| 555 (BAT) | 0 (0) | 190 (100) | 190(100) |
| London Brown/White (BAT) | 0 (0) | 928 (100) | 928 (100) |
| Pallmall (BAT) | 70 (14.2) | 433 (85.8) | 494 (100) |
| Business Royal (Independent Tobacco Inc) | 210 (70.0) | 90 (30.0) | 300 (100) |
| Fine (unknown) | 181 (78.3) | 50 (21.6) | 231 (100) |
| Rothmans Kingsize (BAT) | 29 (1.6) | 1798 (98.4) | 1827 (100) |
| Oris (Oriental General Trading Inc) | 107 (81.1) | 35 (18.9) | 132 (100) |
| Rothmans Royals (BAT) | 99 (86.1) | 20 (13.9) | 115 (100) |
| Gold Seal (China Tobacco) | 85 (91.4) | 8 (8.6) | 93 (100) |
| Tusker (BAT) | 29 (100) | 0 (0) | 29 (100) |
| Others (Fisher, menthol, Cherry etc.) | 61 (50.0) | 61 (50.0) | 122 (100) |
| *P-value** | *<0.001* | | |

Italic values signifies p<=0.05
*P value based on $\chi^2$ or Fisher's exact test.
BAT, British American Tobacco.

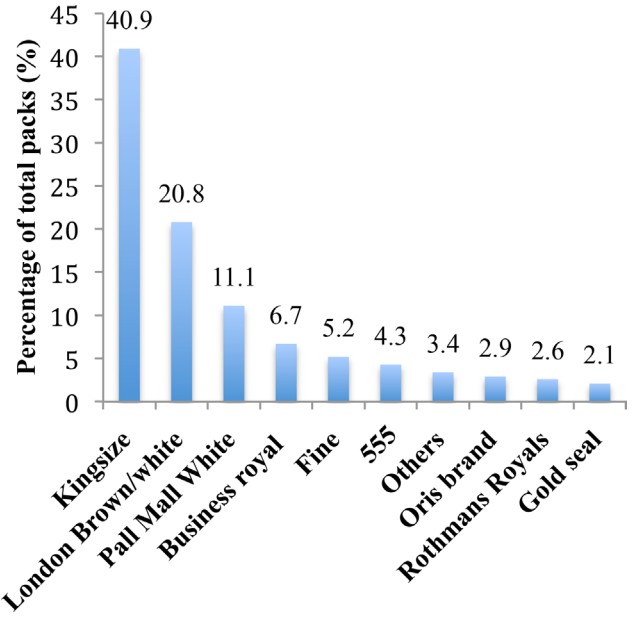

**Figure 2** Cigarette brands sold in Ghana.

and adjusted models, respectively (table 2). Also, the sale of illicit cigarettes was 19.3 and 67.2 odds higher in border towns as compared with non-border towns in both the adjusted and unadjusted models, respectively. The middle and coastal country zones had lower odds of illicit cigarette sales than the northern zones in both the unadjusted and adjusted regression models, respectively. Also, for every unit increase in price/pack, the odds of illicit cigarette consumption reduce by almost 60%.

Table 3 shows the results of bivariate and multivariate analysis adjusted for 384 vendors that collected packs from single stick sales. After adjusting for the clustering effect of vendors, convenience stores had higher odds of illicit cigarette sales in both the bivariate and multivariate analysis adjusted for vendors. Border towns also had higher odds of illicit in both bivariate and multivariate models

## DISCUSSION

This study found out that close to 20% of the packs collected were illicit of the total 4461 packs. A majority of the illicit packs were reported from Aflao (Ghana-Togo border) (99%) and Tamale (46%). Tamale, although not a border town, is the capital of the Northern region of Ghana and has most of the cigarettes smuggled from Burkina Faso.[8] Close to half of the illicit packs originated from Togo (51%), followed by Nigeria (15%) and then Cote d'Ivoire (10%). The most common brand of cigarettes sold in Ghana was from BAT, including Rothmans Kingsize, London Brown/White and Pall Mall. One out of four illicit packs belonged to Business Royal (Independent Tobacco Company), a fifth were from Fine (unknown company) and about 1 out of 10 were from Oris brand (Oriental General Trading). The most common features identified for classifying packs as illicit were the absence of tax stamps, *for sale in Ghana* sign and warning labels not in English. Adjusted and unadjusted logistic regression models indicated that convenience stores, border towns, northern zone of the country and price/pack had higher odds of illicit cigarette consumption for single stick sales in Ghana.

Our study provides an objective measure and describes the nature of the illicit cigarette market. This plays a critical role in developing comprehensive and effective tobacco control policies, particularly in countries within sub-Saharan Africa such as Ghana, where data on illicit cigarette sales are lacking. Our illicit cigarette estimates from single stick sales of 20% is, however, lower than the estimates of the euromonitor (37% in 2018),[17] which is the only available estimate on illicit cigarettes market in

| Table 2 | Unadjusted and adjusted factors for illicit cigarette sales in Ghana | | | | |
|---|---|---|---|---|---|
| | **Unadjusted** | | | **Adjusted** | |
| **Variable** | **OR** | **95% CI** | | **OR** | **95% CI** |
| Retail shop type | | | | | |
| *Drinking bars* | 1 | | | 1 | |
| *Kiosks* | 0.25 | 0.17 to 0.35 | | 0.52 | 0.28 to 0.96 |
| *Convenience stores* | 1.83 | 1.57 to 2.15 | | 3.47 | 1.92 to 6.26 |
| Country zone | | | | | |
| *Northern* | 1 | | | 1 | |
| *Middle* | 0.03 | 0.01 to 0.05 | | 0.42 | 0.16 to 1.08 |
| *Coastal* | 0.54 | 0.46 to 0.63 | | 0.70 | 0.39 to 1.25 |
| Border/non-border towns | | | | | |
| *Non-border town* | 1 | | | 1 | |
| *Border town* | 19.3 | 16.0 to 23.4 | | 67.2 | (44.2 to 102.2) |
| Pack price | 0.39 (coef=−0.94) | (0.37 to 0.42) (−0.99 to −0.88) | | 0.39 (coef=−0.95) | (0.36 to 0.42) (−1.03 to −0.88) |

**Table 3** Effect of clustering by vendors* on illicit cigarette sales

| Variable | Bivariate | | Multivariate | |
|---|---|---|---|---|
| | OR | 95% CI | OR | 95% CI |
| Retail shop type | | | | |
| *Drinking bars* | 1 | | 1 | |
| *Kiosks* | 0.25 | 0.11 to 0.53 | 0.52 | 0.16 to 1.69 |
| *Convenience stores* | 1.83 | 1.03 to 3.26 | 3.47 | 1.22 to 9.84 |
| Country zone | | | | |
| *Northern* | 1 | | 1 | |
| *Middle* | 0.03 | 0.01 to 0.08 | 0.42 | 0.01 to 2.51 |
| *Coastal* | 0.54 | 0.30 to 0.95 | 0.70 | 0.22 to 2.27 |
| Border/non-border towns | | | | |
| *Non-border town* | 1 | | 1 | |
| *Border town* | 19.3 | 8.80 to 42.40 | 67.2 | (17.62 to 256.41) |
| Pack price | 0.39 (coef=−0.94) | (0.31 to 0.50) (−0.99 to −0.89) | 0.39 (coef=−0.95) | (0.32 to 0.46) (−1.05 to −0.89) |

*Adjusted for the clustering effect of vendors on illicit cigarette sales (n=384).

Ghana. Nevertheless, the euromonitor data are criticised for lack of transparency and their funding source from the TI.[17] The TI is known for quoting high estimates of the illicit market as a means of deterring governments from imposing tobacco tax increases, which contributes to ineffective tobacco control and lost opportunities for the governments to collect more revenue.

There are various methods to assess the extent of illicit tobacco in any country, such as measuring the difference between consumption and tax paid sales (gap analysis), interviewing smokers, examination of littered cigarette packs and econometric modelling.[18] We employed a methodology particularly suitable in countries with single stick sales, similar to methods used in India,[9] Pakistan,[19] Bangladesh[10] and Argentina.[20] Despite a ban on single stick sales, all retailers (100%) sold single sticks, calling for enforcement of the ban. Our estimates of illicit cigarette sales (20%) are also similar to countries with a higher tobacco use prevalence such as Pakistan (18%) and Argentina (14%) that used a similar methodology.[19 20] Despite the lack of estimates of illicit cigarettes from many countries in the African Region, countries such as South Africa, Kenya, The Gambia and Nigeria have available estimates of their illicit market using different methods of estimation. Our estimates were found to be lower than South Africa (with over 30% of the total market being illicit),[21] Nigeria (26%)[22] and Kenya (26%)[23] but higher than the Gambia (8.6%).[24] With the recent ratification of the Protocol in Ghana, and estimates suggesting one out of five cigarette packs to be illicit, there is an urgent need for governments to address this by fully implementing ratified protocol (which has specific requirements to improve traceability of tobacco products and increase tobacco industry accountability).

BAT continues to dominate sales of cigarettes as evidenced by the most common cigarettes sold in Ghana (Rothmans Kingsize, London Brown/White and Pall Mall). This is largely due to the company's long history in Ghana.[25] While the company ceased domestic production in 2006, it remains the dominant importer of cigarettes into the country.[25] There are also very low-priced brands available, such as BAT's Tusker brand (of which all packs were illicit). While, all packs from London Brown/White were found to be licit, about 14% of Pall Mall and 1.6% of Rothmans Kingsize were illicit, demonstrating the possibility of the industry's involvement in illicit trade.[26] Furthermore, the small-scale convenience stores were found to be a major selling point of illicit cigarettes. These are legally operating, widely available settings to the low-income Ghanaian smoker (who prefers to buy single stick) widely available in both rural and urban locations. Convenience stores were also found to have higher odds of illicit cigarette consumption as compared with drinking bars in both the adjusted and unadjusted logistic regression models, indicating that it may be an important predictor of illicit cigarette sales in the country.

Geography was found to play an important role in the illicit cigarette market in Ghana. A third of the packs collected from the northern zone of the country were found to be illicit. According to the Euromonitor,[8] the north of Ghana sees particularly strong illicit trade, with most smuggling from Burkina Faso finding their way to this region into Tamale.[17] This could also be linked to the high smoking prevalence and lower income population in the region as compared with other regions.[27] Similarly, border towns were also found to be strong predictors of illicit cigarette sales. Six out of 10 packs collected from border towns were illicit and almost 100% of the packs

collected from Aflao (Ghana-Togo border), and close to half of the packs from Tamale (large city in Northern Ghana linked to Burkina Faso) were found to be illicit. Border towns have been found to be more vulnerable to the trade of illicit cigarette and tobacco products in Vietnam[28] and Georgia.[29] Our findings reinforce the need for strengthening patrolling and border control in addition to building capacity and training for authorities belonging to customs, police and immigration. The illicit cigarettes originated from Togo (51%), followed by Nigeria (15%) and then Cote d'Ivoire (10%).

In terms of pricing of cigarettes, illicit packs were found to be almost 50% cheaper than licit packs. Africa, in general, lags behind other regions (such as European and the Americas) in implementing strong tobacco tax policies.[1] Close to 90% of the illicit packs were belonged to the low price category (2–7 GHC). Currently, the total excise tax on tobacco products, in Ghana, accounts for only 31.8% of the average retail price.[30] Also, over half of the smuggled cigarettes in the study originated from Togo where a pack of cigarettes is priced at about one USD and is about 0.50 USD in Ghana.[30] The link between tobacco taxation and smuggling has been doubtful and inconsistent.[31] According to a report by the World Bank,[32] taxes and prices have only a limited impact on illicit cigarette market share at country level, contrary to arguments by the tobacco industry. The African region, with low prices and low taxation on tobacco products and high levels of smuggling, provides a good illustration of this observation. This calls for more research to understand the relationship between tobacco taxation and smuggling in Africa.

Our study findings should be considered in the light of some limitations. First, despite the wide geographical dispersion in the three zones of the country (northern, middle and coastal), the representativeness to the country is limited. Also, as data were collected during COVID-19 lockdown period in Ghana and we could not explore other border towns that were planned due to pertaining restrictions at that time. Second, the empty pack collection relies on retailers to provide us with all the empty packs from previous day's single stick sales. It could be possible that some retailers would want to hide the illegal packs, which could underestimate our findings. Nevertheless, retailers were motivated with a monetary incentive, which, to an extent, mitigated this issue.

## CONCLUSION

Our study found a total of 20% illicit packs in the entire sample of packs collected across the eight border and non-border towns/cities in Ghana. This study provides valuable information for policymakers and law enforcement in the region and bringing to light the inadequacy of the current monitoring and regulatory activities of the FDA and customs. Our findings have two important policy implications; first, the regulatory body and the focal point for tobacco control in Ghana (FDA) in collaboration with the customs, police and immigration, should strengthen the supply chain control and market surveillance at retail points in the towns and cities, particularly those close to the Ghana-Togo and Ghana-Burkina Faso border in the northern and coastal zones of the country, aside from border monitoring and transportation tracing. Second, with the introduction of Tax Stamp Policy since March 2018, Ghana should also consider the implementation of a supply chain control that resembles a track and trace system (like Kenya), independent of any industry influence to effectively monitor the illicit market.

**Author affiliations**
[1]Faculty of Social Sciences, Tampere University, Tampere, Finland
[2]University of Cape Town Research Unit on the Economics of Excisable Products, Rondebosch, Western Cape, South Africa
[3]The University of Edinburgh College of Medicine and Veterinary Medicine, Edinburgh, UK
[4]Usher institute, University of Edinburgh, Edinburgh, UK
[5]University of Bath, Bath, UK
[6]Research & Development Division-Ghana Health Service, Ghana Health Service, Accra, Greater Accra, Ghana
[7]Research Division, Ministry of Health/Ghana Health Service, Accra, Ghana
[8]Tobacco Control and Substance Abuse, Food and Drug Authority, Accra, Ghana
[9]Department for Health, University of Bath, Bath, UK
[10]Usher Institute and SPECTRUM Consortium, College of Medicine and Veterinary Medicine, University of Edinburgh, Edinburgh, UK
[11]Department of Global Health, Kwame Nkrumah University of Science and Technology, Kumasi, Ashanti, Ghana

**Acknowledgements** We would also like to thank Portia Boakye, Michael Ababio and Christopher Bekoe for their role and contribution in pack collection and fieldwork. We would also like to thank all the retailers that provided information and packs for this survey.

**Contributors** AS, FD, AG, AnG, TK, HR and EOD contributed to the design, conception, acquisition, analysis, and interpretation of the project and data; the drafting and revision of the manuscript and the approval of the final version to be published. AS and DL contributed to the acquisition of data. LB and EOD contributed to the design and conception of the project. All authors contributed to the approval of the final version to be published. EOD and AS accepts full responsibility for the work and/or the conduct of the study, has access to the data and the decision to publish.

**Funding** This work was supported by the Medical Research Council [grant number MR/P027946/2] with funding from the Global Challenges Research Fund and with additional funding from the University of Edinburgh's Scottish Funding Council Global Challenges Research Fund (GCRF) allocation.

**Map disclaimer** The inclusion of any map (including the depiction of any boundaries therein), or of any geographic or locational reference, does not imply the expression of any opinion whatsoever on the part of BMJ concerning the legal status of any country, territory, jurisdiction or area or of its authorities. Any such expression remains solely that of the relevant source and is not endorsed by BMJ. Maps are provided without any warranty of any kind, either express or implied.

**Competing interests** None declared.

**Patient and public involvement** Patients and/or the public were not involved in the design, or conduct, or reporting, or dissemination plans of this research.

**Patient consent for publication** Consent obtained directly from patient(s)

**Ethics approval** The study protocol was approved by the Committee on Human Research, Publication and Ethics (Reference number: CHRPE/AP/441/18) and the University of Bath's Research Ethics Approval Committee for Health (REACH) (EP 19/20 063). Participants gave informed consent to participate in the study before taking part.

**Provenance and peer review** Not commissioned; externally peer reviewed.

**Data availability statement** Data are available upon reasonable request. The data are owned and shared by the Tobacco Control Capacity Program (TCCP) and the School of Public Health, KNUST, Ghana. Requests for data sharing can be made to artisingh_uk@yahoo.com/arti.singh@tuni.fi.

**ORCID iDs**
Arti Singh http://orcid.org/0000-0002-7460-0119
Fiona Dobbie http://orcid.org/0000-0002-8294-8203
Linda Bauld http://orcid.org/0000-0001-7411-4260

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
