## [Reviewer comments · BMJ Open]

ARTICLE DETAILS

TITLE (PROVISIONAL)	Extent of illicit cigarette market from Single Stick Sales in Ghana: findings from a cross sectional survey
AUTHORS	Singh, Arti; Ross, Hana; Dobbie, Fiona; Gallagher, Allen; Kinnunen, Tarja; Logo, Divine; Boateng, Olivia A.; Gilmore, Anna; Bauld, Linda; Owusu-Dabo, Ellis

VERSION 1 – REVIEW

REVIEWER	Barnett, Adrian Queensland University of Technology, Institute of Health and Biomedical Innovation
REVIEW RETURNED	12-Apr-2022

GENERAL COMMENTS	I enjoyed reading this interesting paper about cigarette sales in Ghana. The approach to quantify illicit sales used a good study design. The study was well described. The consent percentage was very high. I fully understand the need for a financial compensation for retailers' time, but is there any concern that the incentive could skew the results? Was it set to the right amount? Could it have encouraged them to add additional packs as the compensation was commensurate with the number of packs provided. The 95% confidence interval for the percentage of illicit cigarettes should be given at line 194 and in the abstract. Did the data collectors include people with local knowledge? Was there any training of the data collectors? Was there any validation of the data collection, for example, randomly re-checking a 1km data collection route to ensure that all vendors were captured? The unit of analysis is the cigarette pack. There is clustering within vendor as there are multiple packs per vendor, meaning these results are likely to be correlated. Hence a slightly different statistical approach is needed where the vendor is added as a cluster or a random effect. This is easily achievable in R, for example using the <code>`glmer`</code> function from the <code>`lme4`</code> package. The code will be almost identical to the logistic regression model, but with the intercept for vendor. There was no EQUATOR checklist completed. STROBE would probably be best. Minor comment - Is there likely to be any difference between data collected on weekdays and weekends? Or by the time of day?
--

	 - Percents over 10% can generally be rounded to whole figures and I think that would aid understanding here with no impact on accuracy, see doi: 10.1136/archdischild-2014-307149. Similarly, the large odds ratios (above 10) in table 2 could be rounded to 1 decimal place. - Line 150, the sample size calculation is missing the margin of error. - Line 171, it feels like a waste to categorise the data that was originally collected on a continuous scale, see https://www.jstor.org/stable/3702100?seq=1 - Line 183, a causal diagram would be useful to investigate which variables were potential confounders. - Line 188, please check if this is the recommended text when there is no patient or public involvement. - Figure 1 had a relatively low resolution in the PDF. - Figure 2, the last bar was missing a label. It wasn't clear if the bars were in any particular order. I recommend ordering from high to low to quickly show the top selling brands. - Table 1. Zero p-values cannot exist, better to say <0.001, or whatever threshold they are below.
--	---

REVIEWER	Welding, Kevin Johns Hopkins University Bloomberg School of Public Health, Institute for Global Tobacco Control
REVIEW RETURNED	24-May-2022

GENERAL COMMENTS	Major comments: Throughout I think you need to be very clear that you are presenting results from single stick sales and not the whole cigarette market. You haven't made an argument that the single stick results are the vast majority of cigarette sales in Ghana or that they are representative of the whole market. You repeatedly state that the results, even in the title, are estimating the extent of illicit cigarettes sales. For example, you state "the most common brand of cigarettes sold in Ghana is...". Related to the comment above, it is unclear when you are referring to results that you coded from the empty packs from single stick sales and results from the retailer survey that presumably gathered information beyond single stick sales. It is not clear to the reader what exactly is covered in the retailer survey. The incentive strategy might have unintended consequences. Paying based on the number of empty packs provided seems like it would incentivize them providing additional empty packs. They could be providing empty packs from previous days to get more incentive. Were they aware of the incentive based on the number of packs before providing the packs? This is a potential limitation. The collected price information is not clear. Is this collected in the retailer survey about the single stick packs? Was price/pack adjusted by stick count? Not adjusting for stick count might cause problems. The same brand variant in 10 and 20 stick varieties could end up in different price categories. Is the pack price the stick price multiplied by the number of sticks or are these packs prices? The country of origin information is not clear. How was country of origin determined? Based on a country of manufacture listed? Based on the language or HWL on the pack? I didn't see this detailed. Was this able to be determined for all packs?
--

	Illicit indicators: Your illicit classification includes no tax stamps, no text/pictorial warnings, no inscription, and HWLs not in English, but then you report illicit characteristics in line 224 as no tax stamps, no inscription, and HWLs in English. Are packs supposed to have HWLs in English or not? It is confusing. It might be worth talking about what the legal HWL looks like in Ghana. Is it pictorial or text? What language is it in? That could help put some of the indicators in context. You introduce the idea of packs not conforming with current labeling requirements approved by the FDA as illicit, but previously you have defined illicit as packs where duties have not been paid. Current labeling requirements don't necessarily mean that duties weren't paid previously. It is important not to confound illegal and illicit. The discussion includes information not presented in the results. For example, in line 261, you mention information gained by retailers like daily retail volume and pack characteristics of the cheapest brand sold, but do not present them in the paper. You should present the results or remove this information from the methods and discussion. Industry involvement in illicit trade: It is a huge leap from buying illicit single sticks from BAT brands and saying this demonstrates the industry's involvement in illicit trade. Did you do any analysis to see if these were counterfeit packs that copied BAT brands? Is it possible that these were tax paid in another country and purchased and transported by someone who is not part of BAT. We would need to know more information about what indicators caused the BAT packs to be illicit to make any judgements. You present information in the discussion that doesn't follow from the results nor has a citation. For example, in line 319: "Nigerian products are mostly smuggled in via Togo and most products smuggled in from Togo originate from BAT's Nigerian operations, with lower taxes in Nigeria enabling these to be sold at a lower price in Ghana." You have provided no evidence or citation for this claim. How do you know Nigerian products came via Togo? Did something on the pack tell you this? This might be answered by more detail about how you identified country of origin. Line 322: You are discussing HWL language, but have presented no results on this matter. You need to add that to the results or remove from the discussion. You say cigarettes from Togo are less affordable as compared to Ghana. You have provided no citation for this claim. The paper could benefit from some copyediting. Specific comments: Abstract: Are the objectives met by looking at single cigarette sales? If not, then adjust the objectives to be about illicit single stick cigarette sales instead of the whole illicit market. Period missing at the end of the sentence for Setting. The outcome measures can be clearer that this is empty packs from vendors selling single sticks and not empty packs found as litter.
--	---

	Introduction: Line 90: I believe single stick should be two separate words. Line 95: You already introduced the WHO abbreviation so you don't need to write it out again. Line 104: Are the unregulated and unlicensed vendors just the bars or also kiosks and street vendors? It is hard to tell what you are referring to here. Line 108: What/who are implementers? The government agencies that have to implement and enforce the policies? Will the reader know this? Line 111: Use "such as" or "including" but not both. Line 112: "there have been no" instead of "there have no"? Line 120: Should there be a comma after "method"? Methods: You say 4 major cities and list 5 cities. You said 4 border towns and list 3. Check the spelling of retailer. Sometimes you use retailer and sometimes retailer. "Up to 10 smaller geographical areas" Were there different numbers of areas for different cities? Line 145: It might be more appropriate to call the money an incentive instead of a reward. Illicit classification includes the absence of authentic tax stamps. Were there occasions where the tax stamp might have fallen off? Did you look at the correlation between the 4 different measures to become illicit? This could be a useful piece of information. Results: Line 192: the word "by" is not needed. Line 193: the sentence has 100% listed twice. You report the drinking bar illicit percentage twice in this section. Check if Pallmall is one word or two. In line 224, there is no mention of the prevalence of packs without text/pictorial HWLs. Table 2 doesn't add much additional information than Table 1. The ordering of the illicit percentage is clear from the percentages in Table 1. What else is in the adjusted model? Discussion:
--	---

	The first paragraph of the discussion might not be necessary since it provides no information that isn't already in the results section. Line 260: Sub-Saharan Africa should be written out. Line 264: The comparison of single stick illicit percentage and the Euromonitor full market illicit is misleading. The difference could also be attributed to the difference in years. Line 281: Should this be "the Gambia"? Line 285: It should be "1 out of 5" and not "1 out 5". Line 304: "Significant predictor" seems like a strong claim. It is true that if you buy a single stick in a convenience store it is more likely to be illicit than in other store types studied. If you shut down convenience stores would the illicit trade decrease or would the prevalence at different vendor types increase? Hard to say if supply or demand is driving the difference. Line 314: it is very confusing to mention Tamale which you categorize as a non-border town in the sentence about border town illicit. You say Tamale is "linked to Burkina Faso" but do not explain what this means and this is the first time the link is mentioned. Explaining why Tamale is an outlier of non-border towns seems important. Line 337: Seems odd to state that price levels do not predict levels of illicit trade, but then say price levels need to be coordinated to decrease illicit trade. You define FDA on line 367, but use it 4 times before that. Conclusion: The second policy implication doesn't seem to come directly from your results. You haven't presented much about the differences in tax or price levels across countries.
--	--

VERSION 1 – AUTHOR RESPONSE

Comments from editor		
1	Please add a section entitled 'Strengths and limitations of this study' (immediately after the abstract). This section should contain up to five short bullet points, no longer than one sentence each, that relate specifically to the methods.	We thank the editor for this comment. We have now added the section on the strengths and limitations' as required by the journal right after the abstract.
2	-Please remove the "What this study adds" section - this is not part of the BMJ	This section has now been removed.

	Open format.	
3	-Along with your revised manuscript, please include a copy of the STROBE checklist indicating the page/line numbers of your manuscript where the relevant information can be found (https://strobe-statement.org)	We have now added a completed STROBE checklist as recommended by the editor as part of the submission.
Comments for reviewer 1		
4	I enjoyed reading this interesting paper about cigarette sales in Ghana. The approach to quantify illicit sales used a good study design. The study was well described.	We are thankful to the reviewer for this positive comment of our article.
5	The consent percentage was very high. I fully understand the need for a financial compensation for retailers' time, but is there any concern that the incentive could skew the results? Was it set to the right amount? Could it have encouraged them to add additional packs, as the compensation was commensurate with the number of packs provided?	Thank you for your comment on the compensation. The incentive that was given is almost a standard amount that researchers give for most studies here in Ghana. We did not think it would be appropriate to give anything less than this. As the amount given was minimal we do not expect retailers to provide extra packs based on the amount.
6	The 95% confidence interval for the percentage of illicit cigarettes should be given at line 194 and in the abstract.	The 95% CI has now been added to the abstract and in the results section as suggested by reviewer.
5	Did the data collectors include people with local knowledge? Was there any training of the data collectors? Was there any validation of the data collection, for example, randomly re-checking a 1km data collection route to ensure that all vendors were captured?	We had a 3-day training prior to data collection on data collection and the pack survey method. A study coordinator always accompanied data collectors to verify the daily data collection process randomly. We also had a good mix of research assistants who were also fluent in the local dialects of the regions that were surveyed and were familiar with the local context.
6	The unit of analysis is the cigarette pack. There is clustering within vendor as there are multiple packs per vendor, meaning these results are likely to be correlated. Hence a slightly different statistical approach is needed where the vendor is	We thank the reviewer for this comment. We were not able to conduct the analysis based on the clustering effect of the vendor. As vendor provided packs ranging from 4-12 packs. We therefore conducted analysis on the clustering effect by the eight cities.

	added as a cluster or a random effect. This is easily achievable in R, for example using the `glmer` function from the `lme4` package. The code will be almost identical to the logistic regression model, but with the intercept for vendor.	However, the cluster effect of the cities on illicit sales produced very wide confidence intervals indicating a lot of uncertainty. We have attached the output from STATA (figure 1 on this document) for the reviewer's information. In light of this, we would like to maintain our current analysis and put this as a potential limitation of the study if the reviewer agrees.
7	There was no EQUATOR checklist completed. STROBE would probably be best.	We have now included a completed STROBE checklist as part of the submission.
8	Is there likely to be any difference between data collected on weekdays and weekends? Or by the time of day?	We thank you for this observation. Smoking habits in Ghana are not culturally acceptable and open as in other countries. Smoking habit does not depend on day of the week or time like alcohol consumption but rather driven by addiction so we did not observe any striking pattern related to day and time of pack collection as the average number of packs collected were similar in both weekdays and weekends (Tobacco In Ghana, Euromonitor 2019).
	Percents over 10% can generally be rounded to whole figures and I think that would aide understanding here with no impact on accuracy, see doi: 10.1136/archdischild-2014-307149. Similarity, the large odds ratios (above 10) in table 2 could be rounded to 1 decimal place.	This has now been addressed throughout the paper. We changed the percentages about 10% to whole numbers in the tax but we maintained them to one decimal place in the tables (1&2).
	Line 150, the sample size calculation is missing the margin of error.	The margin of error has now been added to the statement on sample size calculation.
	- Line 171, it feels like a waste to categorise the data that was originally collected on a continuous scale, see https://www.jstor.org/stable/3702100?seq=1	We have addressed this comment by categorising the price variable for the measure of association but we used the continuous price variable for the regression part of the analysis.
	Line 183, a causal diagram would be useful to investigate which variables were potential confounders.	As suggested by the reviewer, we have included a causal diagram as figure 2.

	Line 188, please check if this is the recommended text when there is no patient or public involvement.	We inserted the wording as such as it was the recommended text in the submission process.
	Figure 1 had a relatively low resolution in the PDF.	The resolution of the figure has now been improved.
	Figure 2, the last bar was missing a label. It wasn't clear if the bars were in any particular order. I recommend ordering from high to low to quickly show the top selling brands.	We thank you for this helpful observation. We have now modified the figure in descending order as suggested.
	Table 1. Zero p-values cannot exist, better to say <0.001, or whatever threshold they are below.	Thank you for the comment on the p-values. We have now adjusted the p-values to <0.001 as suggested by reviewer.
Comments from reviewer 2		
1	Throughout I think you need to be very clear that you are presenting results from single stick sales and not the whole cigarette market. You haven't made an argument that the single stick results are the vast majority of cigarette sales in Ghana or that they are representative of the whole market. You repeatedly state that the results, even in the title, are estimating the extent of illicit cigarettes sales. For example, you state "the most common brand of cigarettes sold in Ghana is...".	We are thankful to the reviewer for this comment. We have made various edits to the relevant sections including the abstract to make it clear that we are presenting results of the illicit cigarette market from single stick sales.
2	Related to the comment above, it is unclear when you are referring to results that you coded from the empty packs from single stick sales and results from the retailer survey that presumably gathered information beyond single stick sales. It is not clear to the reader what exactly is covered in the retailer survey.	We thank the reviewer for this comment. This study covered the results obtained from the pack survey from single stick sales and not our retailer survey. The findings of the retailer survey are part of another paper that we are currently working on.
3	The incentive strategy might have unintended consequences. Paying based on the number of empty packs provided seems like it would incentivize them providing additional empty packs. They	The incentive that was given is almost a standard amount that researchers give for most studies here in Ghana. We did not think it would be appropriate to give any thing less than this. As the amount given was minimal

	could be providing empty packs from previous days to get more incentive. Were they aware of the incentive based on the number of packs before providing the packs? This is a potential limitation.	we do not expect retailers to provide extra packs based on the amount. Retailors were also not pre-informed of higher incentive based on number of packs. All retailers were given an average incentive in the range of \$7-10.
4	The collected price information is not clear. Is this collected in the retailer survey about the single stick packs? Was price/pack adjusted by stick count? Not adjusting for stick count might cause problems. The same brand variant in 10 and 20 stick varieties could end up in different price categories. Is the pack price the stick price multiplied by the number of sticks or are these packs prices?	The prices provided in the paper are price/pack of 10-stick or 20-stick packs and not price/stick in a pack. We have now made this clear in the paper. We have inserted a statement on line 167-168 as “Pack prices were recorded for each of the 10 or 20 stick packs” .
5	The country of origin information is not clear. How was country of origin determined? Based on a country of manufacture listed? Based on the language or HWL on the pack? I didn't see this detailed. Was this able to be determined for all packs?	The country of origin was detected based on the inscription on the packs such as for sale in Togo or Nigeria etc. It was not based on the language on the health warnings. We have included a sentence in the data analysis section (line 244-245) as “based on the inscription on the packs on sale restricted to respective country”
6	Illicit indicators: Your illicit classification includes no tax stamps, no text/pictorial warnings, no inscription, and HWLs not in English, but then you report illicit characteristics in line 224 as no tax stamps, no inscription, and HWLs in English. Are packs supposed to have HWLs in English or not? It is confusing. It might be worth talking about what the legal HWL looks like in Ghana. Is it pictorial or text? What language is it in? That could help put some of the indicators in context.	We apologise for this omission. The statement has now been rephrased as “Majority of the illicit packs were characterized by absence of tax stamps (94.3%), ‘for sale in Ghana’ sign (92.2%) and warning labels not in in English (77.3%)” on lines 262-263. We have also inserted the current warning label format for Ghana warnings on line 188-191 “Current pack warnings in Ghana are required to be a combined picture and text health warning in English to cover 50% of the front principal display area and 60% of the back principal display area of the pack, positioned in the lower portion” .
7	You introduce the idea of packs not conforming with current labeling requirements approved by the FDA as illicit, but previously you have defined illicit	We thank the reviewer for this comment. We have removed the statement on the “duties not paid” on line 211 as it is not part of the

	as packs where duties have not been paid. Current labeling requirements don't necessarily mean that duties weren't paid previously. It is important not to confound illegal and illicit.	criteria we used to classify packs as illicit.
8	The discussion includes information not presented in the results. For example, in line 261, you mention information gained by retailers like daily retail volume and pack characteristics of the cheapest brand sold, but do not present them in the paper. You should present the results or remove this information from the methods and discussion.	As suggested by the reviewer, we have removed this section of the text from the discussion, as it is part of our second paper findings specifically on the retailer survey.
9	Industry involvement in illicit trade: It is a huge leap from buying illicit single sticks from [Note: Removed by editor at acceptance. Please see final version of manuscript.] brands and saying this demonstrates the industry's involvement in illicit trade. Did you do any analysis to see if these were counterfeit packs that copied [Note: Removed by editor at acceptance. Please see final version of manuscript.] brands? Is it possible that these were tax paid in another country and purchased and transported by someone who is not part of [Note: Removed by editor at acceptance. Please see final version of manuscript.]. We would need to know more information about what indicators caused the [Note: Removed by editor at acceptance. Please see final version of manuscript.] packs to be illicit to make any judgments.	We did not do any forensic analysis to check for counterfeit packs. We included this claim only based on the literature reviewed from the OCCRP. According to the OCCRP documents, [Note: Removed by editor at acceptance. Please see final version of manuscript.](see: https://www.occrp.org/en/loosetobacco/british-american-tobacco-fights-dirty-in-west-africa). We have included this literature source in the discussion to make this point clearer.
10	You present information in the discussion that doesn't follow from the results nor has a citation. For example, in line 319: "Nigerian products are mostly smuggled in via Togo and most products smuggled in from Togo originate from [Note: Removed by editor at acceptance. Please see final version of manuscript.], with lower taxes in Nigeria enabling these to be sold at a lower price in Ghana." You have provided no	We thank you for this comment. We have included the citation for the information on cigarette prices in Togo and Nigerian operations. We have deleted the section on the HWL in the discussion section as suggested by the reviewer since we did not present results on that section. We have also included the reference for the claim on cigarettes being less affordable in Togo as

	evidence or citation for this claim. How do you know Nigerian products came via Togo? Did something on the pack tell you this? This might be answered by more detail about how you identified country of origin. Line 322: You are discussing HWL language, but have presented no results on this matter. You need to add that to the results or remove from the discussion. You say cigarettes from Togo are less affordable as compared to Ghana. You have provided no citation for this claim.	compared to Ghana.
11	Abstract: Are the objectives met by looking at single cigarette sales? If not, then adjust the objectives to be about illicit single stick cigarette sales instead of the whole illicit market. Period missing at the end of the sentence for Setting. The outcome measures can be clearer that this is empty packs from vendors selling single sticks and not empty packs found as litter.	We thank the reviewer for this comment. We have reworded the objective in the abstract to read as “This study aims to measure the extent of illicit cigarette consumption from single stick sales, to determine the nature and types of illicit cigarettes present, and to identify the factors associated with illicit cigarettes consumption in Ghana”. We have added a period in the setting sentence. We have modified the outcome measure to read as “estimates of the share of illicit cigarette packs in the total cigarette sales from vendors selling single cigarettes in Ghana”
12	Introduction: Line 90: I believe single stick should be two separate words.	We have revised it to single sticks.

Line 95: You already introduced the WHO abbreviation so you don't need to write it out again. Line 104: Are the unregulated and unlicensed vendors just the bars or also kiosks and street vendors? It is hard to tell what you are referring to here. Line 108: What/who are implementers? The government agencies that have to implement and enforce the policies? Will the reader know this? Line 111: Use "such as" or "including" but not both. Line 112: "there have been no" instead of "there have no"?	This has been revised accordingly. We have revised this statement to now read as "Tobacco products including cigarettes in Ghana is mostly sold at unlicensed and unregulated points such as traditional grocery retailers (also known as convenience or provision stores), street vendors, kiosks and drinking bars" at line 102-105. We have removed the word implementers and kept it to only policy makers. We have revised this statement on line 111-112 to read as follows " In light of the tobacco industry's use of illicit trade to oppose tobacco control measures such as tax increases" We have corrected this statement accordingly.
--	--

	Line 120: Should there be a comma after “method”?	We have added a comma after “method”
13	Methods: You say 4 major cities and list 5 cities. You said 4 border towns and list 3. Check the spelling of retailer. Sometimes you use retailer and sometimes retailer. “Up to 10 smaller geographical areas” Were there different numbers of areas for different cities? Line 145: It might be more appropriate to call the money an incentive instead of a reward. Illicit classification includes the absence of authentic tax stamps. Were there occasions where the tax stamp might have fallen off?	Apologies for this, we have addressed this omission and replaced this section with the following; “five major cities in Ghana (Accra, Tamale, Kumasi, Takoradi and Bolgatanga) and three border towns (Aflao, Paga/Hamele and Elubu)”from line 126-127. We have replaced the spelling of retailer and made it consistent throughout the paper. We thank the reviewer for this observation. We have now clarified this statement to read as “10 smaller geographical areas”. The word reward in line 145 has now been replaced with the word incentive as suggested by reviewer. The tax stamps in Ghana are firmly fixed to the packs and very difficult to manipulate as observed during the pack examination. According to the criteria for classification of illicit packs, even the absence on one feature makes a pack illicit. We are also not sure if the reviewer means association or correlation

	Did you look at the correlation between the 4 different measures to become illicit? This could be a useful piece of information.	here. We would be grateful for more clarity on this comment.
14	Results: Line 192: the word “by” is not needed. Line 193: the sentence has 100% listed twice. You report the drinking bar illicit percentage twice in this section. Check if Pallmall is one word or two. In line 224, there is no mention of the prevalence of packs without text/pictorial HWLs. Table 2 doesn't add much additional information than Table 1. The ordering of the illicit percentage is clear from the percentages in Table 1. What else is in the adjusted model?	The word 'by' has been removed. We have deleted the additional 100% in the text. The additional percentage for drinking bars has been removed. Pallmall has now been replaced with the corrected form - Pall Mall. We have now inserted the prevalence of packs without text/pictorial HWLs into the results section as requested by the reviewer. We thank the reviewer for this comment, however, we would want to maintain both table 1 and 2 as table 1 shows the association (based on chi square) and table 2 shows the strength of association (from regression analysis).
15	Discussion: The first paragraph of the discussion might not be necessary since it provides no information that isn't already in the results	The first paragraph is a summary of our key findings that is also part of the STROBE checklist requirement for observational studies. For this matter, we would like to request the reviewer to enable us to maintain

section. Line 260: Sub-Saharan Africa should be written out. Line 264: The comparison of single stick illicit percentage and the Euromonitor full market illicit is misleading. The difference could also be attributed to the difference in years. Line 281: Should this be “the Gambia”? Line 285: It should be “1 out of 5” and not “1 out 5”. Line 304: “Significant predictor” seems like a strong claim. It is true that if you buy a single stick in a convenience store it is more likely to be illicit than in other store types studied. If you shut down convenience stores would the illicit trade decrease or would the prevalence at different vendor types increase? Hard to say if supply or demand is driving the difference. Line 314: it is very confusing to mention Tamale which you categorize as a non-border town in the sentence about border town illicit. You say Tamale is “linked to Burkina Faso” but do not explain what this means and this is the first time the link is mentioned. Explaining why Tamale is an outlier of non-border towns seems important.	this. SSA has now been written in full in line 260. We understand that the comparison of our findings with Euromonitor is not justified but that was done due to the fact that the only available estimate for Ghana currently is from the Euromonitor. We have included a statement to make this clear (line 370-371) We have now included “the” before Gambia. We have corrected this statement to one out of five. We have revised this statement to now read as “ indicating that it may be an important predictor of illicit cigarette sales in the country”from 330-331. We have clarified this statement in the discussion section and added at line 333-334 “Tamale is the capital of the Northern region of Ghana with most cigarettes smuggled from Burkina Faso”. We have also included the citation for this claim.
--	--

	Line 337: Seems odd to state that price levels do not predict levels of illicit trade, but then say price levels need to be coordinated to decrease illicit trade. Y You define FDA on line 367, but use it 4 times before that.	We have edited the section to keep the discussion on taxation and illicit cigarettes minimal, as our study was not on tobacco taxes. We have revised this section (line 459-466) to read as; “The link between tobacco taxation and smuggling has been doubtful and inconsistent. According to a report by the World Bank, taxes and prices have only a limited impact on illicit cigarette market share at country level (ref), contrary to arguments by the tobacco industry. The African region has both low prices and low taxation on tobacco products and high levels of smuggling provides a good illustration of this observation. This calls for more research to understand the relationship between tobacco taxation and smuggling in Africa”. We have now included a statement from line 160-166 to define FDA earlier in the methods section – “A conservative definition to classify an illicit cigarette pack (packs on which appropriate duties have not been paid) in Ghana according the Food and Drugs Authority (FDA), the regulatory body and the focal point for tobacco control in Ghana”
16	Conclusion: The second policy implication doesn't seem to come directly from your results. You haven't presented much about the differences in tax or price levels across countries.	We thank the reviewer for this observation. We have removed the policy implication related to tax and price levels in the conclusion section and maintained the other two policy implications.

--	--	--

VERSION 2 – REVIEW

REVIEWER	Barnett, Adrian Queensland University of Technology, Institute of Health and Biomedical Innovation
REVIEW RETURNED	13-Aug-2022

GENERAL COMMENTS	The authors have answered all my queries except for the point about clustering. These data have been gathered in two clusters: vendors and cities. So to analyse the data without adjusting for clustering will inflate the sample size and potentially create type I errors. As the authors noted in their response when they used a cluster for city: "cluster effect of the cities on illicit sales produced very wide confidence intervals indicating a lot of uncertainty." This increased uncertainty likely comes from the correct adjustment for correlated data. These results are more likely to be unbiased and need to be the main results used, after also additionally adding vendor as a cluster. It is not sufficient to list the lack of adjustment for clustering as a limitation. Fisher and Chi-squared are not appropriate for these data. Minor comments - Abstract, the confidence interval does not need to be to two decimal places (also line 204) - "No patients and/or the public were not involved in the design" double negative
--

VERSION 2 – AUTHOR RESPONSE

Comments for reviewer 1		
4	These data have been gathered in two clusters: vendors and cities. So to analyse the data without adjusting for clustering could give very biased results. Not accounting for clustering will inflate the sample size and potentially create type I errors. As the authors noted in their response when they used a cluster for city: "cluster effect of the cities on illicit sales produced very wide confidence intervals indicating a lot of uncertainty." This increased uncertainty likely comes from the correct adjustment for correlated data. These results are more likely to be unbiased and need to be the main results used, after also additionally adding vendor as a cluster. It is not sufficient to list the lack of adjustment for clustering as a limitation. Fisher and Chi-squared are not appropriate	We thank the reviewer for this insightful comment. We have included a new table (table 3) presenting results on the clustering effect of 384 vendors on illicit cigarette sales. We however did not see any major differences after adjusting for vendors and our results were almost similar to that obtained after regression analysis. Our study was informed by previous literature (see citation below please) that used similar methodology and hence we did not consider the effect of vendors in an LMIC setting. We collected the packs based on the different cities (border and non-border) and not by vendor, as previous literature has also not indicated the effects of vendors on illicit cigarette sales in LMICs and also from the nature of the cigarette sales in Ghana. 1. John RM, Ross H. Illicit cigarette sales in Indian cities: findings from a

	for these data.	retail survey. Tob Control 2018; 27:684–688 2. Pizarro ME, Giacobone G, Shammah C, et al. Illicit tobacco trade: empty pack survey in eight Argentinean cities Tobacco Control 2022;31: 623-629. Although, we are happy to include the results of the clustering effect, we would want to include it as a supplementary material if the reviewer agrees.
5	Minor comments  - Abstract, the confidence interval does not need to be to two decimal places (also line 204) - "No patients and/or the public were not involved in the design" double negative 	The Confidence interval in the abstract has been edited to one decimal place. This sentence has now been edited to read "No patients or the public were involved in the design, or conduct, or reporting, or dissemination plans of our research".